# An Emerging Visible-Light Organic–Inorganic Hybrid Perovskite for Photocatalytic Applications

**DOI:** 10.3390/nano10010115

**Published:** 2020-01-07

**Authors:** Bianca-Maria Bresolin, Samia Ben Hammouda, Mika Sillanpää

**Affiliations:** 1Laboratory of Green Chemistry, School of Engineering Science, Lappeenranta University of Technology, Sammonkatu 12, 50130 Mikkeli, Finland; samiabenhammouda@gmail.com (S.B.H.); Mika.Sillanpaa@lut.fi (M.S.); 2Department of Civil and Environmental Engineering, Florida International University, Miami, FL 33174, USA

**Keywords:** halide perovskite, photocatalysis, visible-light, Rhodamine B, oxidation

## Abstract

The development of visible-light active photocatalysts is a current challenge especially energy and environmental-related fields. Herein, methylammonium lead iodide perovskite (MAIPb) was chosen as the novel semiconductor material for its ability of absorbing visible-light. An easily reproducible and efficient method was employed to synthesize the as-mentioned material. The sample was characterized by various techniques and has been used as visible-light photocatalyst for degradation of two model pollutants: rhodamine B (RhB) and methylene-blue (MB). The photo-degradation of RhB was found to achieve about 65% after 180 min of treatment. Moreover, the efficiency was enhanced to 100% by assisting the process with a small amount of H_2_O_2_. The visible-light activity of the photocatalyst was attributed to its ability to absorb light as well as to enhance separation of photogenerated carriers. The main outcome of the present work is the investigation of a hybrid perovskite as photocatalyst for wastewater treatment.

## 1. Introduction

Nowadays, environment pollution and energy related issues captured the attention of new century researchers [1,2,3,4,5]. In particular, accelerated release of pollutants because of a combination of growing population and a rapid industrial development have dramatically increased the water pollution in many parts of the world. On the other hand, it is equally urgent to answer the increasing energy demand and mitigate the negative effect of global warming by means of renewable energy sources. Thus, efficient and eco-friendly methods for the degradation of organic pollutants based on renewable energy source, such as solar light, have become an imperative task worldwide [6,7,8,9,10,11].

Heterogeneous photocatalysis consists in the dispersion of a solid material, usually a semiconductor, that when irradiated at appropriated wavelengths is capable to generate highly reactive oxygen species (ROS) which can degrade organic pollutants [12,13]. Photocatalysis main advantages are: the room temperature operation, the utilization of clean and renewable solar light as the driving force and any production of hazardous residues after mineralization to align with the “zero” waste scheme for industries [14,15].

Recently, hybrid organic–inorganic halide perovskites (HOIPs) have gain a lot of attention, especially in photovoltaics, because of their remarkable properties. It was 2009 when for the first time Miyasaka and his colleagues employed hybrid perovskites in photovoltaic devices [16]. Then, the studies of the HOIPs have stunned the research community with their remarkable performance and rapid progress [17].

Perovskite general formula is ABX_3_. HOIP A-site is occupied by an organic cation, B-site by a metal od group IVA in a divalent oxidation state and X-site by a halogen anion [18]. As reported in previous studies, the electronic properties of the mentioned perovskites is mainly governed by the B-X bonds [19]. Electronic properties are fundamental in the understanding of heterogeneous photocatalysis [20].

Herein, focusing on the compositional, structural, optical, and charges transportation properties, we investigated this class of materials as promising candidate for photocatalytic applications [18,21,22,23,24,25]. First, the advantageous properties are a favorable mobility of the photogenerated charges, a reduced surface recombination and long electron–hole diffusion length because of the strong defect tolerance, the shallow point defects and the benign grain boundary. Second, these materials are known to own an enhanced visible-light shift absorption ability and suitable band gap [23,24]. Moreover, they can be produced by low cost solution processes [26].

According to literature, lead-based HOIPs (MAIPb) has achieved the best efficiency among all the studied hybrid perovksites [27,28,29,30,31]. In MAIPb, A-site is occupied by methyl ammonium cation (CH_3_NH_3_^+^), the B-site by lead cation (Pb^2+^), and the X-site by iodine anion (I^−^) (Figure 1) [32,33].

Herein, we propose to determine the feasibility of MAIPb, as one of the most promising HOIPs, as visible-light photocatalyst for the degradation of some dyes having different chemical structures. In particular we investigate the photocatalytic degradation on rhodamine B (RhB, fluorone dye) and methylene blue (MB, thiazine dye) because these dyes are extensively used in industries and medicines [34,35]. Moreover, the effect of key operating conditions on degradation efficiency were studied: catalyst loading, addition of hydrogen peroxide, radiation intensity, solution pH, solution temperature, pollutant initial concentration, and potential recycling test.

## 2. Experimental

### 2.1. Materials

Methylamine (CH_3_NH_2_, 33 wt% in ethanol), hydriodic acid distilled (HI 57 wt% in water), diethyl ether (DE purity ≥ 99.8%), lead (II) iodide (PbI_2_ purity 99%), γ-butyrolactone (GBL purity ≥ 99%) were purchased from Sigma Aldrich (Darmstadt, Germany) and used as received. The target dye pollutants RhB, was obtained from Sigma Aldrich (Darmstadt, Germany).

### 2.2. Photo-Catalyst Synthesis

Hybrid organic-inorganic perovskite was prepared with a one-step, solution-processed method as described in previous literature report [18]. CH_3_NH_2_ (11.39 mL, 0.09 mol) and HI (10 mL, 0.08 mol) were stirred for 2 h in an ice bath kept at 0 °C to synthesize the precursor, CH_3_NH_3_I. The solution was evaporated at 50 °C and the solid was washed three times with DE and dried at 50 °C on a hot plate. The CH_3_NH_3_I (0.39 g) and PbI_2_ (1.16 g) were mixed in GBL (10 mL). Finally, the sample was dried at 60 °C for 6 h until the solution was completely evaporated. Before performing photo-catalytic oxidation process, the catalyst was washed several time with deionized water. It should be mentioned that methylammonium was selected as precursor because it is most widely used as A-site cation since its radius appeared to be the more suitable resulting in low packing symmetry and high band gap [36,37]. In comparison to other elements of group IV, Pb was selected because of its performance and stability [18,38,39,40]. In particular, along group IV from Pb to Ge, it was previously reported a decrease in stability of the divalent oxidation state and a consequent decrease in band gap value combined with a reduced inert electron pair effects [41]. Among the halogens, iodide was selected for its higher efficiency compared to other elements [16]. Moreover, in the periodic table iodide lies close to Pb, thus, they result in more stable structure by sharing similar covalent character [18]. However, we must notice that many factors remain not entirely understood. Moreover, some barriers are still to overcome as stability and toxicity in large-scale implementation.

### 2.3. Photo-Catalyst Characterization

The X-ray powder diffraction (XRD) spectrum of the catalyst was recorded by PANalytical instrument with the empyrean program (PANalytical, Cambridge, UK) with Co-Kα (λ = 1.7809 Å) as the radiation source, 40 kV generator voltage and 40 mA tube current. The diffraction angle (2θ) ranged from 20°and 80° with intervals of 0.05°. The sample functional groups were characterized by Fourier transform infrared spectra (FT-IR) (Bruker, Solna, Sweden) in the region from 400 to 4000 cm^−1^ at room temperature using Horiba FT-730 FT-IR spectrometer. The microstructure and morphology of the material were defined using scanning electron microscope (SEM) Hitachi SU3500 (Chiyoda, Tokyo, Japan). Energy dispersive spectroscopy (EDS) (Thermo Scientific, Waltham, MA, USA) detected the elemental composition of the pure hybrid organic-inorganic perovskite. The surface composition and the electronic states of elements in the valence-band region were determined by ESCALAB 250 X-ray photoelectron spectroscopy (XPS) (ThermoFisher Scientific, Waltham, MA, USA) with Al-Kα (1486.6 eV) as the X-ray source. Absorption spectra were measured with a PerkinElmer Lambda 1050 spectrophotometer (UV-vis) (PerkinElmer, Waltham, MA, USA) to establish the absorption spectrum and band gap of the sample.

### 2.4. Procedure for Photo-Catalysis

The visible-light photocatalytic efficiency was evaluated based on the degradation of RhB. All experiments were carried out in Pyrex vessels (100 mL) with 50 mL of RhB (20 mg∙L^−1^). Specified amount of reaction mixture was withdrawn at regular time intervals and analyzed with UV-vis spectrophotometer at emission wavelength of 554 nm [42]. The efficiency of RhB removal was determined as follow:Removal efficiency % = *C*/*C*_0_(1)
where *C*_0_ is the initial concentration of RhB and *C* is the measured concentration at the time of withdrawal [43,44]. Electron spin resonance (ESR) technique with proper spin traps was used to determine the presence of reactive oxygen species (ROS). TEMP (2,2,6,6-tetramethylpiperidine) was used as spin trap for singlet oxygen and DMSO (dimethyl sulfoxide) for superoxide and hydroxyl radicals [45,46]. The specifics of the visible-light device, used in the current research, are reported in the Appendix A (Appendix A and Appendix A).

## 3. Results and Discussion

### 3.1. Photo-Catalyst Characterization

The morphology of the material was investigated with SEM, the results, shown in Figure 2A,B, suggest an aggregation of nanoparticles with hexagonal shape domains with nanometers size. The specific morphology of the crystal lattice is mainly influenced by the synthesizing temperature and may affect the optical, electrical, and transmission properties of the material, as confirmed in the study of Li et al. [47].

In order to access the sorption behavior of these materials in aqueous phase, N_2_ sorption can provide some useful information for the characterization and evaluation of the performance of the photocatalyst [48]. As indicated by the analysis in Appendix A, the sample showed type III according to IUPAC classification.

Appendix A shows EDS spectra of the sample. The analysis confirms the presence of C, N, O, Pb, and I. The ratio C:N:I:Pb was found to be 4.06:0.58:42.33:49.62. Lower signals for carbon and nitrogen can be assigned to their lighter atomic weights. XRD pattern of the sample is presented in Figure 3A. Hexagonal crystal system was mainly detected with space groups *P3m1*. Dominant diffraction peaks at 2θ = 14.7, 30.2, 40, 46.26, and 53.01° were assigned respectively to the (002), (012), (104), (110), and (106) facets of the hexagonal crystalline structure. Peaks at 14.7 and 30.2° were also be indexed to (110) and (220) facets of the tetragonal structure of perovskite according to literature [49]. It should be noted the diffraction peaks of PbI_2_, assigned at 2θ equals to 12.8°. The miller indexes (h, k, l) recorded suggested more than one preferred crystal orientation in our samples.

Appendix A displays FT-IR spectrum of the synthesized organo-halide perovskite. The sample showed broad vibrations N-H from 2800 to 3350 cm^−1^, the characteristic features of hydrogen bonds overlapped the C-H vibrations signs. The peaks at 1450 cm^−1^ and around 650–750 cm^−1^ belong to the organic cation vibrations since the Pb-I and Pb-I-Pb appeared in very lower energy [50,51]. Peaks displayed at 1500 and 956 cm^−1^ an be respectively assigned to N-Pb-I stretching mode and Pb-I-NH bending. The wide bend around 3100 cm^−1^ was assigned to CH-NH stretching vibration [52].

The optical properties were further investigated in terms of light absorption capability because the absorption of light energy is one of the key of photocatalytic processes. Hybrid organic–inorganic perovskite achieves an optical absorbance across the entire visible spectrum as highlighted by Dualeh [53]. Carrier diffusion lengths was found to reach up to 100 nm for both electrons and holes in MAIPb via transient photo-luminescence measurements [54,55]. A nearly instantaneous charge generation and dissociations of balanced free charge carriers with high mobility has been observed, and the charges were proved to remain in that state for up to tens of microseconds [56]. From previous literature, it was found that the electronic levels for hybrid perovskites consist of an antibonding hybrid state between the Pb-s and I-p and a non-bonding hybrid state between the Pb-p and I-p orbitals corresponding to highest occupied and lowest unoccupied molecular orbitals, respectively [57]. The electronic properties were not influenced by organic fraction. In particular, Frost et al. showed that VB transition is primarily affected by the ionization potential of halogen ions contribution [58].

In Figure 3B the optical band gap of the perovskite was calculated. From extrapolation of the linear part of the Tauc plot (Kubelka–Munk theory), the optical gap was estimated to be 1.58 eV, which is in close agreement with previous reports [18,21,59,60,61,62].

XPS measurements were performed in order to investigate the chemical bonding states of the element in the envisaged catalyst Figure 4A. According to Navas et al. [18], peaks at 143 and 138.1 eV can be assigned to Pb 4f (Figure 5B); peaks around 412 eV, showed in Figure 4B, were assigned to Pb 4d_5/2_. The bigger peaks can be associated with the Pb component in the halide hybrid perovskite structure, while the smaller to metallic Pb probability decomposed from PbI during the synthesis [63]. Peak corresponding to 401 eV peak were assigned to N1s Figure 4D. In accordance with the studies performed by Chen et al. [64], N state may vary and the associated peaks can be found at different BE. Different peaks positions were found in a range of 396–404 eV in agreement with Nakamura and Mrowetz et al. [65]. Conforming to the study of Navas et al. [18], the peaks shown in Figure 4C belong to I 3d_3/2_ and I 3d_5/2_. It was further shown that the spectrum shows well separated spin–orbit components, separation of around 11.4–11.5 eV was recorded as typical evidence of the presence of I^−0^ [18]. Figure 4E shows peak belongs to C1s around 285 eV. Shen et al. [54], in their interesting research on hybrid organic-inorganic perovskite for solar cell application, assigned this peak to the methyl group. The conclusion obtained here agrees well with that reported by previous literature confirming the achievement of the synthesis processes [18,63,64,65,66].

### 3.2. Photocatalytic Activity

Among the persistent contaminants, organic dye molecules are toxic and their uncontrolled discharge from various industries into the water can have a huge impact on the environment [67]. In our study, the photocatalytic activity of the synthesized nano-catalyst was examined on RhB removal, which is considered as one of the most abundant dyes in the textile industries effluents and commonly chosen as model pollutant for photocatalytic treatment [68,69]. The photocatalytic performance of investigated material was evaluated as the decrease of the relative concentrations of RhB (*C*/*C*_0_) plotted over time in different conditions. The removal efficiency achieved by photolysis was found to be negligible. This fact suggests that the chosen pollutant owns excellent photo-stability, as highlighted by Drexhage et al. and Beija et al. [70,71]. Control experiment in dark conditions was evaluated. The results showed moderate affinity between the halide perovskite and RhB molecules in terms of adsorption in darkness. The result are in accordance with the low surface area measured by BET analysis. As expected, significant improvement on RhB removal efficiency was observed during the photocatalytic experiments Figure 5A. After 3 h of irradiation, the concentration of RhB greatly decreased with respect to the initial concentration, proving the activity of the as-synthesized photocatalyst. The UV-vis spectra indicate that the main absorbance peak was reduced as a function of irradiation time and the dye molecules were decolorized. On the other hand, the peak position was found to be invariable and the diminishing intensity suggested that the fused aromatic ring structures and dye chromophores were destroyed (Appendix A). Kibombo et al. achieved similar results during their researches on optimization of photocatalysts for persistent organic pollutant remediation in wastewater management [72]. In their work it was deeply explained how the reactive oxygen species attack the auxochromic groups, induce N-de-ethylation of the alkyl amine group and how photogenerated holes can degrade both RhB suspended molecules and N-de-ethylated products. As depicted in Figure 5, the removal efficiency appeared at the very first interval (15 min), this is in accordance with the ROS generation that is higher at the earlier step of irradiation [73,74,75].

The potential of the as-prepared material in photodegradation of a different organic compound was further investigated. In particular, methylene blue (MB) was chosen as the target contaminant. Methylene blue could be successfully removed by the assisted photocatalytic reaction after 60 min under visible-light irradiation. The results were compared with blank experiments to demonstrate the photocatalytic nature of the reaction. The results and the comparison are shown in Appendix A (Appendix A).

The photocatalytic activity of the as-prepared nanoparticles showed higher photocatalytic efficiency for MB dye compared to RhB. The differences in the recorded efficiencies can be attributed to the chemical structures of the organic dyes and the nature of the functional groups present on their surfaces.

### 3.3. Effect of H_2_O_2_ on the Photocatalysis Treatment

Many techniques have been applied to reduce the effect of recombination of charges and to enhance the heterogenous photocatalysis performance. Among these techniques, the assistance of external electron acceptor such as hydrogen peroxide (H_2_O_2_) in the photocatalytic process has gained more and more attention. The effect of H_2_O_2_ on photocatalytic oxidation of RhB in aqueous suspensions of the as-synthesized material was investigated. Various concentrations of oxidant were used. Test without the presence of a photocatalyst was performed. In addition, the photocatalytic degradation of RhB was found to follow the pseudo first-order reaction model:ln(*C*/*C*_0_) = −*kt*(2)

The degradation rate constant *k* and the correlation coefficient of the curve *R*^2^ were obtained using regression analysis. The value of *R*^2^ were higher than 0.92, thus it was assumed that the regression line fits well with the data (Table 1).

The reaction rate increased with H_2_O_2_ dosages. For the highest concentration of oxidant (10^−3^ mol∙L^−1^), the kinetic rate was found to be almost 25 times higher than the lowest concentration and 5 times higher than the average concentration. For practical application and considering the cost of hydrogen peroxide, 10^−4^ mol∙L^−1^ was considered as the optimal value. The combination of halide perovskite and H_2_O_2_ under visible-light illumination was found to greatly enhance the degradation rates of RhB. When H_2_O_2_ concentration increases, more hydroxyl radicals are produced thus the oxidation rate increases. ROS were considered as dominant mechanism in the photocatalytic process. The first hypothesis is a direct photolysis of H_2_O_2_ by visible light that may generate free radicals at a wavelength of 405 nm [76]. A second minor mechanism proposed by Ollis et al. [77] and Ilisz et al. [78] suggested that H_2_O_2_ may partially contribute to the rate enhancement of photo-catalytic process behaving as an electron acceptor. According to these theories, H_2_O_2_ cannot only generate ·OH but also as electron acceptor, reduce the electrons-holes recombination increasing the photocatalytic efficiency. On the other hand Dionysiou et al. [79] in their studies on assisted-H_2_O_2_-photocatalysis showed that high concentrations of hydrogen peroxide may decrease the degradation rates because of the consumption of hydroxyl radicals.

### 3.4. Effect of Catalyst Loading

The effect of catalyst load on the ability to remove RhB in aqueous solution is shown in Figure 6A. The results suggest that the removal performance increased with the catalyst load up to 0.5 g∙L^−1^ and decreased when the load is higher. This is in agreement with the case observed in heterogeneous photo-catalysis reaction. This behavior can be rationalized both in terms of availability of active sites on material surface and light penetration of photo-activating light into the system. The availability of active sites increased with catalyst loading, but on contrary the light penetration and, hence, the photo-activated volume of particles decreased [80]. Moreover, higher amount of catalyst may induce the deactivation of particles by collision with ground state molecules reducing the rate of reaction [81]. The trade-off of these effects was studied by considering also the organic contaminant concentration.

### 3.5. Effect of Initial Concentration of RhB

The effect of RhB initial concentration is an important parameter to consider [82]. Figure 6B depicts the effect of RhB initial concentration on its removal. The result reveals that the increase of the RhB concentration decreases the removal, corresponding to those from literature [1].

At higher RhB concentration, the generation of radicals on the surface of catalyst may be reduced by the competition of the active sites covered by RhB ions. Moreover, with the increase in the concentration the photons may be intercepted before they can reach the catalyst surface, decreasing the absorption of photons by the catalyst [83]. Higher concentration of RhB may also cause aggregation and even surface dimerization and have consequentially an effect on the degradation rates [42].

### 3.6. Effect of Initial pH

The pH of the dye solution was altered by adding incremental amounts of either dilute HCl or diluted NaOH in order to study the effect of pH on dye removal. Previously, it was confirmed that none of the salts used had any effect on the dye spectra in the absence of light. The solution was subjected to irradiation and change in absorbance value was noted.

The removal rate was found to increase in acidic media as shown in Figure 6C. The photolytic dye degradation appeared to be the best at pH 3 and it decreased when pH was increased. The results implied that in alkaline medium new oxidizing species, such as hydroperoxy anion can be formed. The new species can react with both the reactive oxygen species such as hydroxyl radicals as well as H_2_O_2_ molecules. This can consequently lower the dye contaminant removal rate. Future studies will be required to clarify the effect of pH on dye discoloration.

### 3.7. Effect of Temperature on H_2_O_2_-Assisted Photo-Catalysis

According to Wang et al. [84], temperature is another parameter that affects the heterogenous photo-catalysis. Therefore, in this study, 25 °C, 35 °C, 45 °C were selected to examine the effect of temperature on photo-catalysis under visible light irradiation. As the temperature increased from 25 to 45 °C, the first-order rate constant k1 increased almost 40% (Table 2). This behavior was associated to a decrease in the viscosity and to an enhanced diffusion of the sorbate molecule [85].

The Arrhenius equation was used to determine the activation energy as follows:*K* = *A**exp(−*Ea*/R*T*)(3)
where *K* is the constant rate that controls the entire process, *A* is the Arrhenius constant, *T* the solution temperature in *K*, *Ea* apparent activation energy (kJ∙mol^−1^), and R the ideal gas constant 0.0083 kJ mol^−1^∙K^−1^. The data are fitted using a linear regression (R^2^ 0.9935). From the Arrhenius-type plot (Figure 7). *Ea* value was calculated as 36.96 kJ∙mol^−1^. Mcheik and El Jamal found similar result in their study on removal of RhB with persulfate and iron activation [86]. The reaction appeared to be activated also at room temperature and proceeded with relatively low energy barrier.

### 3.8. Recyclability of the H_2_O_2_-Assisted Photo-Catalysis System

Recyclability of the photocatalyst represents one of the most important advantages of a heterogeneous application. Thus, the recyclability of the synthesized material was evaluated by using H_2_O_2_ to activate the process for multiple cycles. Figure 8 shows three cycles of the RhB removal using the H_2_O_2_-hybrid organic-inorganic perovskite system irradiated under visible light. It can be seen that after 3 cycles, the system showed a stable and effective catalytic activity on the removal of the selected dye, and the activity loss was negligible. RhB degradation efficiency showed slight decrease from 93% to 80% after 120 min of the third treatment. The results obtained may be caused by active sites saturation. Moreover, the recycle was performed in series, thus a slight decrease in photocatalyst content should be considered. It must be mentioned that the main aim of the former study is to investigate the potential of HOIPs in photocatalytic processes. Further development on material and process technology should be applied.

### 3.9. Active Species and Possible Mechanism

In a typical photocatalytic application, when a semiconductor is irradiated with equivalent or greater light-energy, the electrons (e^−^) in the valence band (VB) are excited into the conduction band (CB) leaving holes (h^+^) in the VB. The photo-generated electrons and holes trigger the redox reaction. When the bottoms of the CB is below the reduction potential of H^+^ to H_2_ (0 V vs. NHE), and the tops of the VB must be located more positively than the oxidation potential of H_2_O to O_2_ (1.23 V vs. NHE) both oxidation and reduction sites are created [87]. The electron/hole pairs and reactive oxygen species (ROS), including O_2_·^−^, and ·OH, are widely considered the main active species responsible for photocatalytic degradation of organic contaminants [88,89].

As deeply studied by Han et al. [90], the electron spin resonance (ESR) spin-trap technique confirms the presence of free radicals. DMPO and TEMP were used as spin trap for superoxide or hydroxide radicals anions (O_2_·^−^, ·OH) and singlet oxygen species (^1^O_2_), respectively.

In detail, upon visible light photo-excitation of the mixture of the organo-halide perovskite and diamagnetic 2,2,6,6-tetramethylpiperidine (TEMP), three lines with equal intensities were observed in the recorded spectrum in Figure 9. This indicates the capture of singlet oxygen (^1^O_2_) generated by TEMP, leading to the formation of the TEMPO radical. The irradiation period was set at 5 min, a signal of g = 2.0001 appeared confirming photo-generation of radicals. The time of irradiation increased and the intensity of peaks decreased, after half-hour of irradiation the resulting spectrum is shown in Figure 9A. The decrease in spectrum intensity of peaks suggests that ^1^O_2_ radical generation occurred in the very first intervals of the photo-catalytic process that is mainly due to their nano-second lifetime [46,91,92]. 5,5-dimethylpyrroline N-oxide (DMPO) was utilized as superoxide and hydroxide radical anions (O_2_·^−^, ·OH) spin trap. Four typical peaks appeared in the ESR spectrum revealing the presence of the radicals, g factor was found equal to 1.9985. Later, the sampling period was increased, and the lower peaks were recorded, indicating that radical generation belongs to the initial period of irradiation. Figure 9B shows the radical peaks after 5 min of irradiation. The signal recorded after 30 min shows a decrease in the intensity of peaks implying that no more radicals are present in the solution.

Finally, to evaluate also the effect of RhB in the production of radicals, a solution of equal content (100 μL) of RhB (20 mg∙L^−1^) and DMPO (100 mM) was prepared and irradiated in the presence of photo-catalyst. After an irradiation time of 5 min, the ESR spectrum was recorded revealing the presence of ·OH radicals (Figure 10). Four typical peaks were recorded also in presence of RhB, revealing a potential synergetic effect between photo-catalyst and organic dye in the production of hydroxyl and superoxide species. A mixed solution of RhB and DMPO was also prepared in the absence of photo-catalyst to confirm the absence of the radicals.

From the results described above, it may be concluded that both ^1^O_2_, O_2_·^−^ and ·OH radicals were produced during the visible-light photo-catalyst treatment of RhB [93,94].

The photocatalytic degradation process proceeds through excitation, transportation, and degradation pathways. As highlighted by Yin et al. [19], during the investigation mechanism of photocatalytic degradation of RhB by TiO_2_/Eosin-Y system under visible light, dye molecules transfer electrons onto conduction bands (CB) of catalyst leading to the formation of dye cationic radicals. Then the involved electrons generate a series of active oxygen species such as O_2_·^−^, ·OH, and ^1^O_2_ which are considered to be involved in the organic contaminant degradation. In a similar study performed by Dutta et al. [95], two main mechanisms were proposed to promote dye degradation, one governed by dye sensitization and the other by the photo-catalyst excitation. In the self-sensitized dye degradation, the photo-induced electrons flow from the dyes to photo-catalyst surface as suggested by their potential energy values. In particular, Lv et al. [96], with their respective co-authors, deeply described the direction of the charge flow; the difference in the potential energy between the CBs induces the electrons to transfer from higher energy level of the photo-excited dye to the lower ones of catalyst. On the other hand, visible light excitation of MAIPb structures could also generate holes in the valance band (VB) and electrons in the CB. Egger et al. studied the tunability of VB (ionization potential) and CB (electron affinity) energies by the atomic orbitals of the anions and cations in different organohalide perovskite [97]. Band energy and band gap engineering of these organic–inorganic solids are indeed possible to be controlled by the chemical composition, and iodine presence was found to upshift the VB and generally narrowing the band gap, favorable condition for bleaching organic compound in aqueous solutions.

The CB transported electrons in both the materials may react with the dissolved oxygen in the water to produce a reactive oxygen species, main responsible for the oxidative dye degradation under visible light irradiation. As confirmed by an interesting study on nanosized Bi_2_WO_6_ performed under visible light by Fu et al. [98], the presence of oxygen is responsible for the activation of photo-catalysis process. In their experiments, they confirmed the importance of the presence of dissolved oxygen in the treated solution, since its effect is primarily to act as an efficient e^−^ trap, leading to the generation of reactive oxygen species and preventing the recombination of charges. Furthermore, Dutta et al. highlighted a similar conclusion in their study on ternary nano-composite based on cadium sulphide (CdS), TiO_2_, and graphene oxide. Herein, they proved how generated electrons react with the dissolved oxygen in water to produce a reactive oxidizing agent initially in the specific form of oxygen radical anion O_2_·^-^, responsible for the oxidative dye degradation under visible light irradiation [95].

Based on the previous discussion, a possible mechanism of RhB is depicted in Figure 11. After self-sensitization of RhB and the excitation of organohalide perovskite, separation of charges occurs, and transport of electron is promoted. On the other hand, dissolved oxygen can act as an electron acceptor, and can be reduced by the promoted electron in the conduction band to form a superoxide specie O_2_·^−^ (3). The O_2_·^−^ can subsequently re-oxidize to ^1^O_2_ or, in the presence of water and H_2_O_2_, it can form ·OH. The strong oxidation power of the hole enables a one-electron oxidation step with water to produce a hydroxyl radical ·OH. These radicals are highly ROS, able to oxidize directly organic contaminant. In our study, the generation of O_2_·^−^ and ·OH was confirmed by the ESR spectra by using DMPO as the spin trap reagent [45], instead, TEMP was used to detect singlet oxygen and it proved electrons and holes generation during visible light irradiation [46].

## 4. Conclusions

In conclusion, bare MAIPb were easily synthesized for the photocatalytic degradation of organic dye pollutants. The degradation performance study suggested that RHB was completely degraded after 180 min of treatment assisted by H_2_O_2_-MAIPb system under visible light irradiation. In this work, we have shown that the outstanding optoelectronic properties of MAIPb can be addressed for photocatalytic degradation of organic compounds. The results constitute a significant step forward in the application of hybrid halide perovskite for solar-driven catalytic processes. It is important to mention that the systematic evaluation of the environmental conditions must be deeply studied.

## Figures and Tables

**Figure 1 nanomaterials-10-00115-f001:**
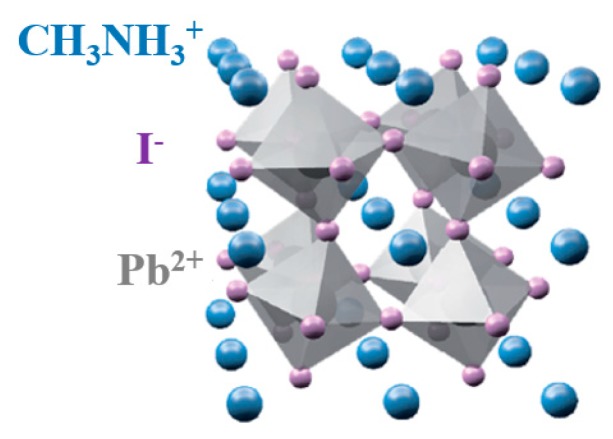
Hybrid organic inorganic perovskite tetragonal structure.

**Figure 2 nanomaterials-10-00115-f002:**
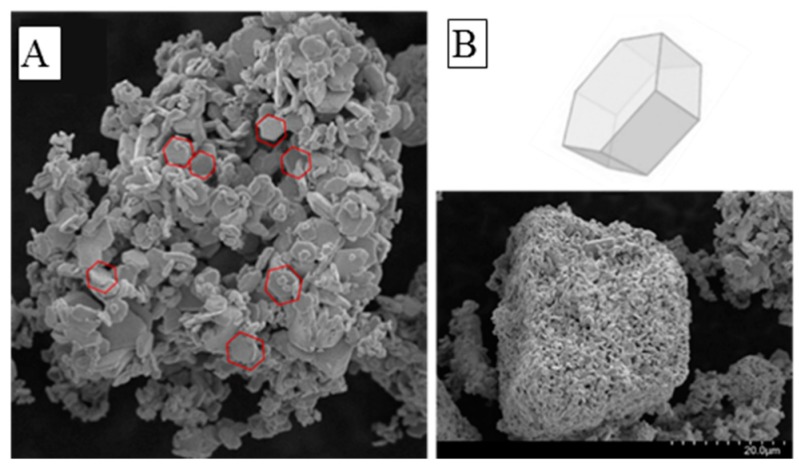
(**A**,**B**) Scanning electron microscopy (SEM) image of the as-prepared MAIPb (methylammonium lead iodide perovskite).

**Figure 3 nanomaterials-10-00115-f003:**
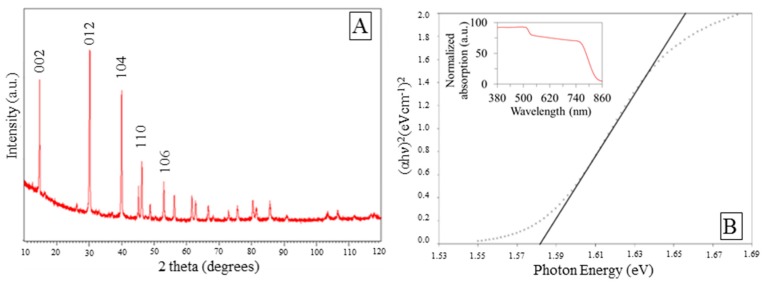
(**A**) X-ray diffraction (XRD) spectrum of the as-prepared MAIPb, (**B**) UV-vis spectrum and Tauc plot of the as prepared MAIPb.

**Figure 4 nanomaterials-10-00115-f004:**
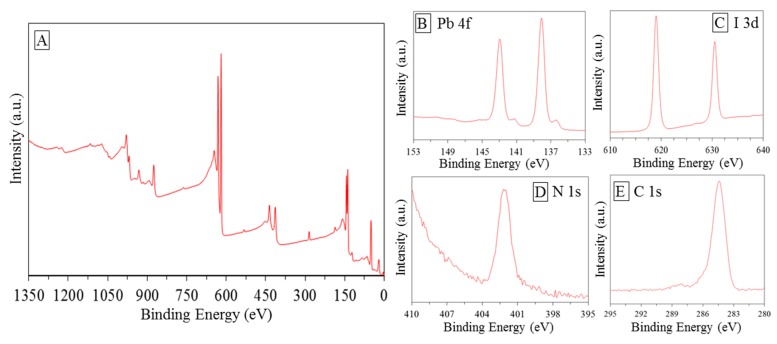
X-ray photoelectron spectroscopy (XPS) spectra of the as-prepared MAIPb: (**A**) general spectrum, (**B**–**E**) zooming on specific binding energy range.

**Figure 5 nanomaterials-10-00115-f005:**
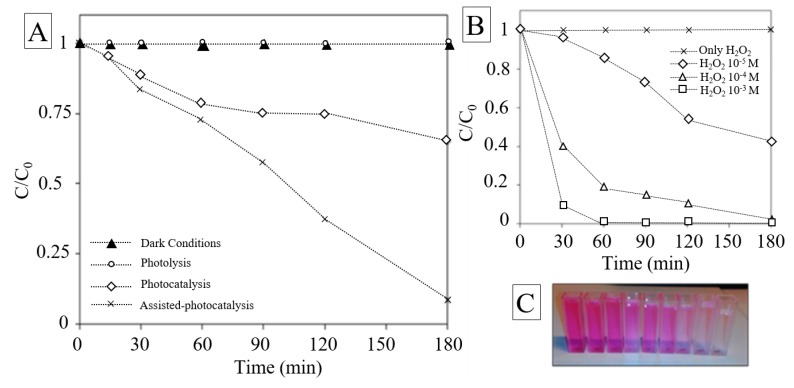
(**A**) Degradation of Rhodamine B (RhB) in dark condition, photolysis, photocatalysis, assisted-photocatalysis; (**B**) degradation of RhB under different H_2_O_2_ concentration conditions 10^−4^, 10^−3^, 10^−5^ mol∙L^−1^; (**C**) picture of color extinction of RhB as function of time.

**Figure 6 nanomaterials-10-00115-f006:**
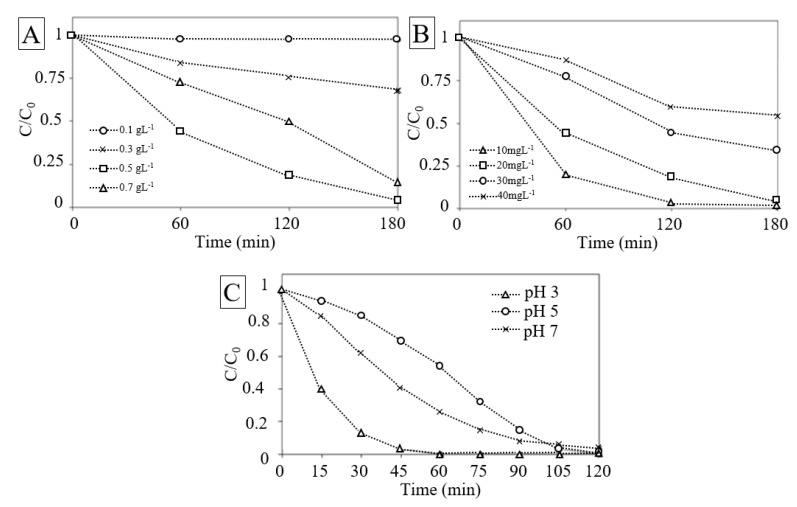
(**A**) Effect of catalyst load. (**B**) Effect of RhB initial concentration. (**C**) Effect of RhB initial pH value.

**Figure 7 nanomaterials-10-00115-f007:**
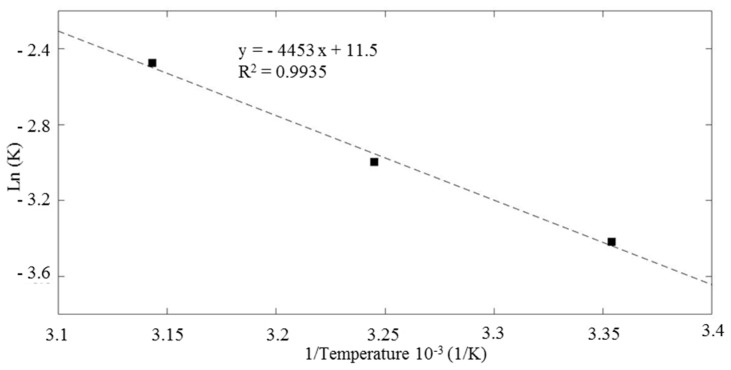
Arrhenius-type plot for the evaluation of the activation energy of the reaction.

**Figure 8 nanomaterials-10-00115-f008:**
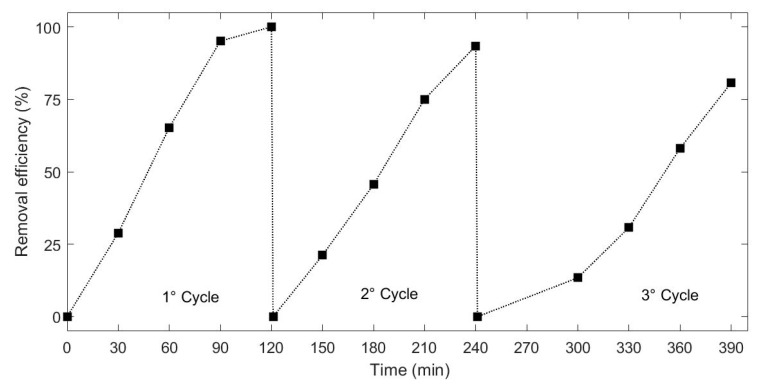
Degradation profile of RhB under assisted visible light photo-catalysis for three cycles.

**Figure 9 nanomaterials-10-00115-f009:**
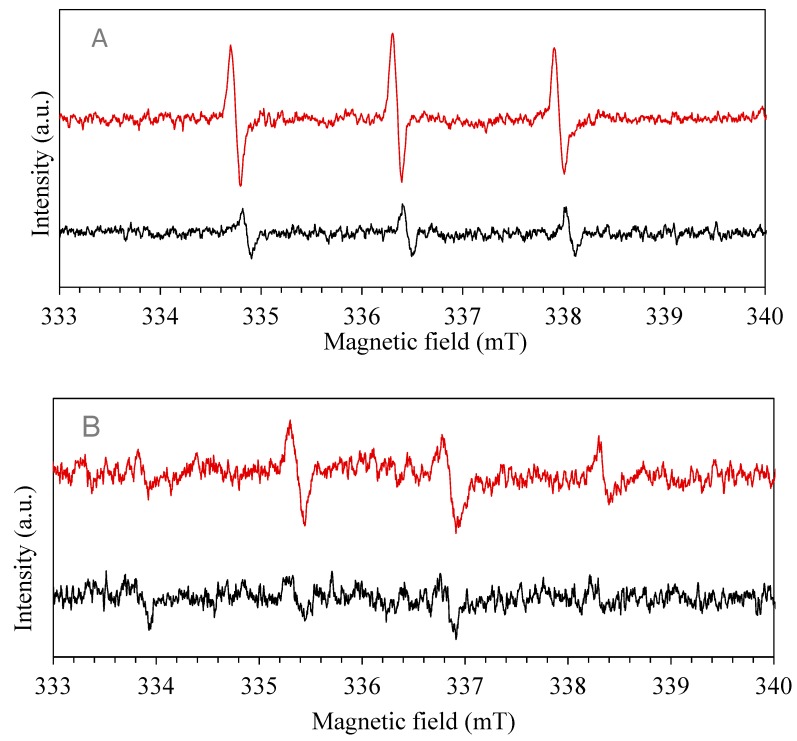
Electron paramagnetic resonance (EPR) spectra using as spin-trap: (**A**) TEMP, red for 5 min, black for 30 min; (**B**) DMPO, red for 5 min, black for 30 min.

**Figure 10 nanomaterials-10-00115-f010:**
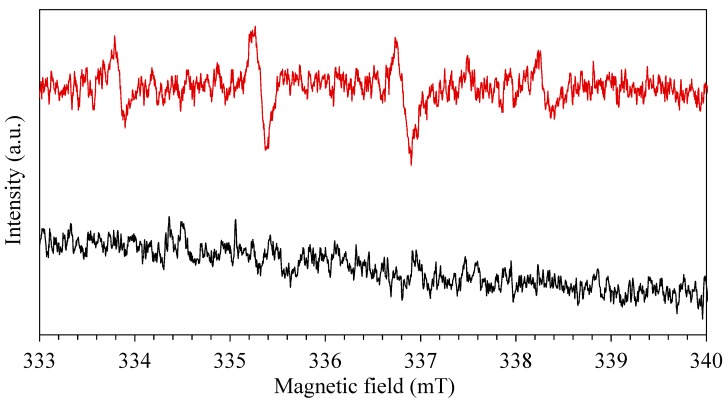
EPR spectra using DMPO as spin-trap in presence of RhB, red with catalyst and black without catalyst.

**Figure 11 nanomaterials-10-00115-f011:**
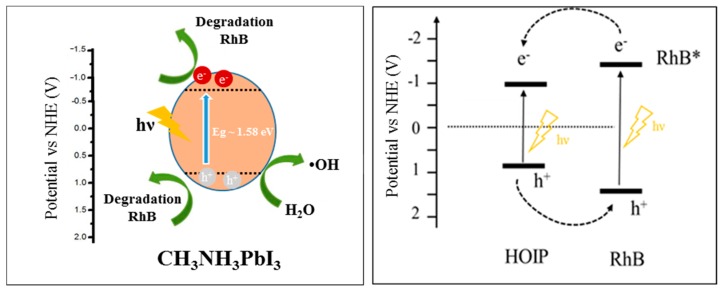
Proposed band gap energy diagram and charge transfer between RhB and photo-catalyst.

**Table 1 nanomaterials-10-00115-t001:** Degradation rate constant k and the correlation coefficient.

Experiment	Rate (s^−1^)	*R* ^2^
Assisted photocatalysis H_2_O_2_ 10^−5^ mol∙L^−1^	0.0045	0.94
Assisted photocatalysis H_2_O_2_ 10^−3^ mol∙L^−1^	0.0215	0.92
Assisted photocatalysis H_2_O_2_ 10^−3^ mol∙L^−1^	0.1087	0.92

**Table 2 nanomaterials-10-00115-t002:** Impact of temperature on the RhB removal kinetic rate under the CH_3_NH_3_PbI_3_/visible irradiation system, experimental conditions RhB: (20 mg∙L^−1^), H_2_O_2_ (10^−4^ M), photo-catalyst (0.5 g∙L^−1^), pH 5.

Temperature (°C)	Kinetic rate (min^−1^)	R^2^
25	0.0328	0.9632
35	0.0499	0.9143
45	0.0840	0.8526

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
