# Peer review of "An Emerging Visible-Light Organic–Inorganic Hybrid Perovskite for Photocatalytic Applications"

_nanomaterials, 2020, doi:10.3390/nano10010115_

Round 1

Reviewer 1 Report

Comments on « An emerging visible-light organic-inorganic hybrid perovskite for photocatalytic applications»

In their contribution, Bresolin et al. report the preparation and characterization of a methylammonium lead iodide perovskite and its application for the photocatalytic degradation of Rhodamine B (RhB) dye in aqueous media. The hybrid perovskite material is first characterized by various spectroscopic techniques and its photoactivity is the evaluated by screening a number of reaction parameters. Finally, based on EPR experiments, the authors discuss a possible mechanism for the photocatalytic activity.

The concept of take advantage of the electronic properties of hybrid perovskites for organic dye photodegradation is certainly appealing, but not highly original since there are already a certain number of papers reported in the literature. For example a recent report of lead-free perovskite photocatalyst (Catal. Sci. Technol., 2017,7, 2753) is not cited in the manuscript. Moreover, in the present contribution the photoactivity of the material seem to be highly dependent on the presence of hydrogen peroxide which somehow decrease the impact of the results.

Overall, the paper is logically presented with fairly well described experiments. The manuscript could however benefit of clearer and more straightforward writing especially when the authors discuss the possible mechanism for photodegradation. This paragraph is very long and it is difficult for the reader to follow the authors reasoning and conclusion.

In my opinion there are some major concerns that need to be addressed before eventual publication in Nanomaterials.

Major concerns

Line 119. The authors claim: “fig 2 show aggregates of nanoparticles with hexagonal domains…” From the provided SEM image it is impossible to make such claim ! Please provide a clear image or modify this sentence. Line 127. “the sample showed type IV to IUPAC classification”. I don’t get how the author can make such claim. Indeed from the provided N2 sorption isotherm (Fig S2), there is no N2 sorption before high relative pressure and certainly no observable hysteresis in the desorption process. It is more likely a type III profile, suggesting a non porous material. It makes no sense to calculate a BET surface area and pore volume. Line 200 “the UV-vis indicate that the main absorbance peak was reduced…” Where is this UV-vis for RhB provided ? Is it supposed to be Fig S5, but this one is for methylene blue ! This brings to another concern in this manuscript. In this abstract and introduction part, the authors claim to investigate the photodegradation of Rhodamine B and Methylene Blue. However, methylene blue is never discussed in the manuscript and only four UV-vis spectra are shown in SI without being referred to in the manuscript. The authors should either remove any mention of methylene blue in the manuscript or provide data from similar experiments done with Rhodamine B. Line 251 typo: “volume of the suspension shrinks decreased” Line 272 and Fig 6c. The authors claim that in acidic solution (pH=3) the photodegradation is retarded, resulting in lower degradation efficiency. However in Fig 6c, the curve corresponding to pH=3 shows the best photodegradation efficiency. Where is the mistake ? Line 309. The author claim that after 3 cycles the system show a stable activity with negligible loss. However the efficiency went from almost 100% to 80% in just 3 cycle. I’m not sure 20% loss can be referred as negligible. Did the author check the stability of their material after photocatalysis ? It would be interesting to provide a comparison of PXRD before and after photocatalysis. Did the author consider the problem of lead leakage upon water treatment with their material? It is appealing to photodegrade water pollutant with photoactive hybrid materials, but it would certainly be problematic to release toxic lead species during this process.

Minor comments

Throughout the whole manuscript there are error messages for captions or references, (Ex line 118 “reference source not found….”) which are not beneficial for overall clarity. In the experimental section, for the catalyst synthesis the authors should provide more a more detailed description by including quantities of reagents and solvent volumes, so that the experiments can be reproduced by others. There is no detail concerning the source of light for the photocatalytic experiments. What kind of lamp, power, wavelength ? Line 137. Typo assigned instead of “assigned” Line 142. Should be figure S4 instead of S3

Author Response

REVIEWER 1

Major concerns:

Line 119. The authors claim: “fig 2 show aggregates of nanoparticles with hexagonal domains…” From the provided SEM image it is impossible to make such claim ! Please provide a clear image or modify this sentence.

Dear reviewer, I provide better image

Line 127. “the sample showed type IV to IUPAC classification”. I don’t get how the author can make such claim. Indeed from the provided N2 sorption isotherm (Fig S2), there is no N2 sorption before high relative pressure and certainly no observable hysteresis in the desorption process. It is more likely a type III profile, suggesting a non porous material. It makes no sense to calculate a BET surface area and pore volume.

Dear reviewer, I agreed with you. I modify the text according your suggestions.

Line 200 “the UV-vis indicate that the main absorbance peak was reduced…” Where is this UV-vis for RhB provided ? Is it supposed to be Fig S5, but this one is for methylene blue ! This brings to another concern in this manuscript. In this abstract and introduction part, the authors claim to investigate the photodegradation of Rhodamine B and Methylene Blue. However, methylene blue is never discussed in the manuscript and only four UV-vis spectra are shown in SI without being referred to in the manuscript. The authors should either remove any mention of methylene blue in the manuscript or provide data from similar experiments done with Rhodamine B.

Dear reviewer, following your suggestion I modify the manuscript to make it more consistent.

Line 251 typo: “volume of the suspension shrinks decreased”

Dear reviewer, I correct the phrase.

Line 272 and Fig 6c. The authors claim that in acidic solution (pH=3) the photodegradation is retarded, resulting in lower degradation efficiency. However in Fig 6c, the curve corresponding to pH=3 shows the best photodegradation efficiency. Where is the mistake ?

Dear reviewer, thank you very much for the highlight. I correct the manuscript in order to make it consistent.

Line 309. The author claim that after 3 cycles the system show a stable activity with negligible loss. However the efficiency went from almost 100% to 80% in just 3 cycle. I’m not sure 20% loss can be referred as negligible. Did the author check the stability of their material after photocatalysis ? It would be interesting to provide a comparison of PXRD before and after photocatalysis. Did the author consider the problem of lead leakage upon water treatment with their material? It is appealing to photodegrade water pollutant with photoactive hybrid materials, but it would certainly be problematic to release toxic lead species during this process.

Dear reviewer, this study has the scope to investigate a new class of materials (HOIPs) for visible-light photocatalysis. According to our purpose, we test a new material applied so far only in solar cells technology: hybrid organic inorganic perovskite HOIP. In particular, CH3NH3PbI3 (MAPbI3) perovskite have attracted intensive attention due to their record power conversion efficiency and low fabrication cost. On the other hand, as reported in many studies, the stability is a critical challenge for this class of materials. It has been shown that MAPbI3 can be degraded to MAI, PbI2, and HI after many hours in contact with water. However, previous studies showed degradation pathways lasting many hours or even days on contrary, we performed minute-scale tests. Indeed, it has been chosen to test the photocatalytic ability of the particles in short reaction time.

Moreover, once performed the photocatalytic test, the solution was filtered to collect the solid material. Further test was performed on the filtrate liquid and the material that might be dissolved inside. No activity was detected. On the contrary, during the recycle test (with the solid residue), activity was recorded during the different runs. Finally, it must be mentioned that this study is a preliminary test on the potential application of HOIP in AOPs.

Minor comments

Throughout the whole manuscript there are error messages for captions or references, (Ex line 118 “reference source not found….”) which are not beneficial for overall clarity.

Dear reviewer, Thank you for the suggestion, I appreciated and corrected the mistakes.

In the experimental section, for the catalyst synthesis the authors should provide more a more detailed description by including quantities of reagents and solvent volumes, so that the experiments can be reproduced by others.

Dear reviewer, thank you vey much for the suggestions. I update the experimental session.

There is no detail concerning the source of light for the photocatalytic experiments. What kind of lamp, power, wavelength ?

Dear reviewer, thank you very much for the suggestions I updated the manuscript and the Supporting information according your advices.

Line 137. Typo assigned instead of “assigned”

Dear Reviewer, thank you very much for the correction.

Line 142. Should be figure S4 instead of S3

Dear Reviewer, thank you very much for the correction.

Reviewer 2 Report

This manuscript shows visible-light organic–inorganic hybrid perovskite for photocatalytic applications. Although results showed excellent performance, but  perovskite photocatalytic properties formed on the interface between RhB and perovskite has insufficient evidence to conclude the mechanism. Although the present work is undoubtedly original, this problem leads to fragmentation of the research results. The reviewer give the authors some comments which are listed as below.

Although CH3NH3PbI3 is sensitive with visible-light, it is possibly degraded itself after the light absorption. The authors need to provide more explanation about its stability.   The authors should provide CH3NH3PbI3 characterizations, such as XRD and UV-Vis before and after Photo-catalyst tests. In Figure 8, the degradation efficiency decreased after the third treatment. The reason need to provide.

Author Response

REVIEWER 2

Although CH3NH3PbI3 is sensitive with visible-light, it is possibly degraded itself after the light absorption. The authors need to provide more explanation about its stability.  

The authors should provide CH3NH3PbI3 characterizations, such as XRD and UV-Vis before and after Photo-catalyst tests.

Dear reviewer, this study has the scope to investigate a new class of materials (HOIPs) for visible-light photocatalysis. According to our purpose, we test a new material applied so far only in solar cells technology: hybrid organic inorganic perovskite HOIP. In particular, CH3NH3PbI3 (MAPbI3) perovskite have attracted intensive attention due to their record power conversion efficiency and low fabrication cost. On the other hand, as reported in many studies, the stability is a critical challenge for this class of materials.

It has been shown that MAPbI3 can be degraded to MAI, PbI2, and HI after many hours in contact with water. However, previous studies showed degradation pathways lasting many hours or even days on contrary we performed minute-scale tests. Indeed, it has been chosen to test the photocatalytic ability of the particles in short reaction time.

Moreover, once performed the photocatalytic test, the solution was filtered to collect the solid material. Further test was performed on the filtrate liquid and the material that might be dissolved inside. No activity was detected. On the contrary, during the recycle test (with the solid residue), activity was recorded during the different runs. Finally, it must be mentioned that this study is a preliminary test on the potential application of HOIP in AOPs.

In Figure 8, the degradation efficiency decreased after the third treatment. The reason need to provide.

Dear reviewer, thank you very much for the suggestion, following your advice I update the manuscript in the former session.

Round 2

Reviewer 1 Report

The authors have taken into account the suggestions made in the previous report.

Academic Editor Notes

The authors did a good job of responding to the concerns of the first reviewer. I think the manuscript could be accepted for publication if the authors add information about their data on methylene blue, which are shown in Figure S6. Although the abstract says that the paper includes the results of the photocatalytic degradation of methylene blue, the only data on methylene blue are shown in Figure S6 with no explanation. This figure needs discussion in the text of the main manuscript to explain the conclusions that the reader should draw from the 4 sections of Figure S6.

The potential of the as prepared material in photodegradation of a different organic compound was further investigated. In particular, Methylene Blue (MB) was chosen as target contaminant. Methylene blue could be successfully removed by the assisted photocatalytic reaction after 60 minutes under visible-light irradiation. The results were compared to blank experiments to demonstrate the photocatalytic nature of the reaction. The results and the comparison are shown in Supporting Information (Figure S6).

Also, how do the results on methylene blue compare with those for RhB?
The photocatalytic activity of the as prepared nanoparticles showed higher photocatalytic efficiency for MB dye compared to RhB. This different efficiencies recorded can be attributed to the chemical structures of the organic dyes and the nature of the functional groups present on their surfaces.

In addition, the authors need to correct the following typographical errors.
Line 150 correct spelling of inrganic to inorganic; correct spelling of achives to achieves.

Thank you for the suggestion and support

Reference source not found.

CORRECT IN THE TEXT